# Enhancing Gradient Boosting Machines with Attention

## Abstract

Gradient boosting machines (GBMs) are a popular machine learning model, well known for their high accuracy and flexibility. Despite significant research in the past two decades improving their accuracy, speed, and robustness, there still lies room for improvement. Notably, with their ability to interpret complex patterns in noisy data. To address this challenge, this paper proposes AMBeRBoost: a novel model that integrates neural attention mechanisms into a GBM, aiming to help the model "focus" on important data and improve its predictive performance on otherwise hard to predict datasets. A series of experiments were performed to evaluate the effects of the attention mechanism, along with the performance of AMBeRBoost against other state-of-the-art models across several publicly available datasets. The results show that AMBeRBoost consistently outperforms the attentionless baseline model on almost all metrics, with results comparable to, and sometimes even exceeding, state-of-the-art models. This research contributes to the continuous improvement and refinement of machine learning models by bridging the gap between GBMs and neural attention mechanisms.

## 1 Introduction

In recent years, machine learning (ML) has revolutionized many fields: healthcare and medicine (Pham et al., 2022; Esteva et al., 2021), chemistry and material sciences (Oliveira and Oliveira, 2022), physics (Pol et al., 2021; Carleo et al., 2019), finance (Ozbayoglu et al., 2020), and many more. In this ML revolution, gradient boosting machines (GBMs) have emerged as some of the most impactful tools. GBMs have shown state-of-the-art results for many practical classification (Rufo et al., 2021; Alzamzami et al., 2020; Cherif and Kortebi, 2019) and regression (Singh et al., 2021; Wang and Mamo, 2020; Zhang and Haghani, 2015), applications. Another influential mechanism that has emerged in the field of ML is attention. From revolutionizing recurrent neural networks (RNNs) (Bahdanau et al., 2016), to serving as the backbone of transformers (Vaswani et al., 2017), attention mechanisms have proven their ability to enhance predictions and interpret complex data, particularly with neural networks. However, despite these many strengths, GBMs still struggle to understand complex relationships in data that is noisy (Glen, 2019). In this paper, a hybrid approach is proposed to help GBMs overcome the challenges pertaining to understanding noisy datasets. Neural attention mechanisms are dynamically implemented into a GBM to help adapt and capture complex patterns in the data. Specifically, an attention mechanism based regression gradient boosting machine, or AMBeRBoost, is proposed. The remainder of this paper is as follows: Section II reviews related literature and relevant concepts. Section III lays out the architecture and implementation of AMBeRBoost. Section IV reviews the experiments, including datasets, hyperparameter selection, and metrics. Sections V presents the results of the experiments. Section VI analyzes the results and their implications. Section VII discusses potential limitations and future research relating to this study. Finally, Section VIII summarizes the findings and the contributions to the field of ML.

## 2 Related Works

### 2.1 Gradient Boosting Machines

Introduced by Friedman (2001), gradient boosting machines are a popular and powerful ensemble learning technique. They work by iteratively fitting weak models (conventionally decision trees) to minimize a

loss function using gradient descent optimization. GMBs have gained significant attention in the field of machine learning for both classification and regression tasks due to their high accuracy, high flexibility, high customizability, and considerable success in various applications (Natekin and Knoll, 2013). As a result of this success, a number of popular GBM models have emerged, including XGBoost (Chen and Guestrin, 2016), LightGBM (Ke et al., 2017), and scikit-learn (noa, c; Pedregosa et al., 2011). In 2015, 17 of the 29 winning solutions to the challenges hosted by the site Kaggle used XGBoost (Chen and Guestrin, 2016).

## 2.2 Attention

In the context of machine learning, attention is a mechanism that dynamically assigns scalar weights to input vectors based on their importance to the task at hand. By training the model on these weights, it can improve both performance and interpretability (Lindsay, 2020). One of the most prominent applications of attention in ML is in neural networks. This was first proposed in Bahdanau et al. (2016), where attention was implemented into a recurrent neural network (RNN) as a way to address the need for the model to compress all of the input information into a fixed-length vector. The addition of attention significantly improved its ability to handle long source sentences. Potentially the most significant use of attention currently, however, is in transformers. Transformers utilize multi-headed self-attention to compute the importance of each word in a sequence relative to each other word, allowing for more nuanced understandings of words and better learning (Vaswani et al., 2017). From March 2020 to March 2022, 70% of arXiv papers on AI mentioned Transformers (Merritt, 2022).

# 3 Methodology

## 3.1 Base Model

A simple GBM regression model was constructed in order to observe the effects of the attention layer in a more direct and controlled manner. This model will be referred to as the "base model." The training steps of the base model are as shown in Algorithm 1:

---
**Algorithm 1:** Training with Base Model

---
**Input:** input $X$ and target $y$
initialize Q as {};
initialize $p_{\text{init}}$ as the mean of $y$;
initialize $y_{\text{pred}}$ with the same shape as $y$ and fill it with $p_{\text{init}}$;
**for** $i=1,...,k$ **do**
    compute $\delta$ as $y_{\text{pred}}$ - $y$;
    train a decision tree regressor with parameters P on $X$ with $-\delta$;
    obtain predictions from the trained tree, $p_{\text{i}}$;
    update $y_{\text{pred}}$ by adding r $\times$ $p_{\text{i}}$;
    append the trained tree to Q;

---

In this algorithm, $k$ is a hyperparameter specifying the number of estimators, $r$ is a hyperparameter specifying the learning rate, the decision tree regressor is obtained from scikit-learn (noa, c; Pedregosa et al., 2011), and $P$ is a set of parameters containing the max depth (a hyperparameter), and a random seed[1] to ensure reproducibility. $Q$ is the ensemble of trees constructing the trained model. The NumPy library (Harris et al., 2020) was used for various numerical and data operations in the training algorithm.

## 3.2 Attention Layer

The attention layer was constructed as a custom TensorFlow layer (Abadi et al., 2015) with two fully connected dense layers. The first layer uses a rectified linear unit (ReLU) activation function to compute attention logits from the features and gradients. The number of hidden units, $h$, is specified as a hyper-

---
[1]All random seeds in this study were set to the number 42.

parameter. The second layer uses a sigmoid activation function to ensure that the attention weights are between 0 and 1. Additionally, random seeds were set to ensure reproducibility. The full implementation of the attention layer is as shown in Algorithm 2:

---

**Algorithm 2:** Attention Layer

---

**Input:** input $X$, gradients $\delta$, and hidden units $h$
expand $\delta$ to match dimension of $X$;
concatenate $X$ and $\delta$;
compute attention logits using a dense layer with $h$ hidden units and ReLU activation;
average logits across concatenated dimension;
normalize logits to $[0, 1]$ using a dense layer with 1 unit and sigmoid activation;
**Output:** attention weights

---

To demonstrate the role of the attention layer, a simple dataset with 500 samples, 2 features, and a noise parameter of 1 was constructed using the make_regression function from the scikit-learn library (noa, e; Pedregosa et al., 2011). Then, the attention weights of one of the features was plotted against the targets ($y$) across training iterations using the Matplotlib library (Hunter, 2007). Figure 1 shows the attention weights at the first iteration, and Figure 2 shows the attention weights at the last iteration.

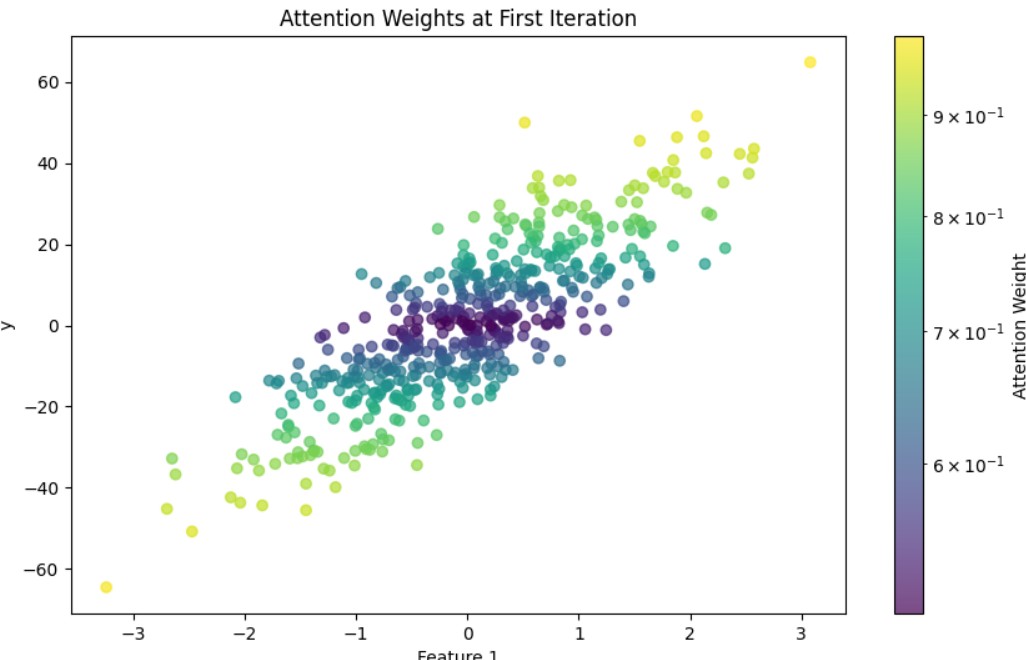

Figure 1: Attention weights of training data visualized at the first training iteration. $y_{pred}$ is initialized as the mean of $y$, so the "difficulty" of predicting points increases as they get further from the mean.

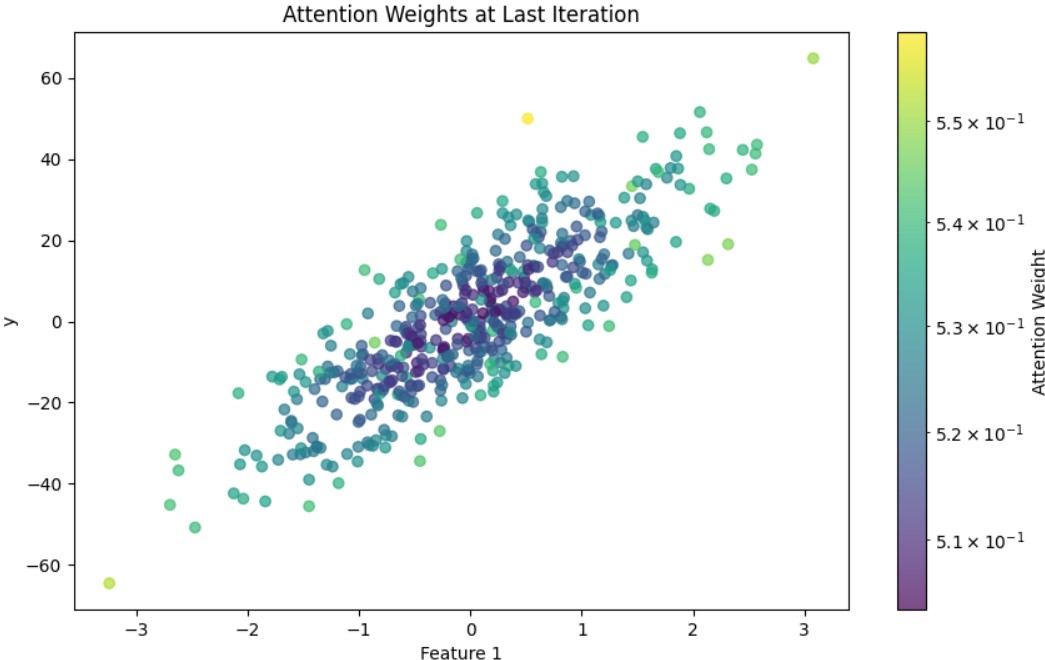

Figure 2: Attention weights of the training data visualized at the last training iteration. Weights are more uniform as the model is better fit to the data. Outliers, like the point near (0.8, 45), are still difficult to predict. Also note that colors in this figure are not numerically equivalent to the same colors in Figure 1.

### 3.3 AMBeRBoost

The final model (AMBeRBoost) was constructed by integrating the attention layer into the base model. The implementation is as shown in Algorithm 3:

---
**Algorithm 3:** Training with AMBeRBoost

---
**Input:** input $X$ and target $y$
initialize Q as {};
initialize $p_{\text{init}}$ as the mean of $y$;
initialize $y_{\text{pred}}$ with the same shape as $y$ and fill it with $p_{\text{init}}$;
**for** $i=1,...,k$ **do**
  compute $\delta$ as $y_{\text{pred}} - y$;
  calculate $w_i$ using Algorithm 2 on $X$, $h$, and $\delta$;
  compute $X_{\text{weighted}}$ with Eqn. 1;
  train a decision tree regressor with P on $X_{\text{weighted}}$ with $-\delta$;
  obtain predictions from the trained tree, $p_i$;
  update $y_{\text{pred}}$ by adding $r \times p_i$;
  append the trained tree to $Q$;
**end**

---

The weighted $X$ values are computed with Equation 1:

$$X_{\text{weighted}} = \left( \left( \frac{1}{\lambda} - 1 \right) \mathbf{1}_n + w_i \right) \mathbf{1}_m \odot \lambda X \tag{1}$$

where:

- $X \in \mathbb{R}^{n \times m}$ is the original input matrix with $n$ samples and $m$ features

- $\lambda$ is a hyperparameter that specifies the influence of the weights on $X$

- $w_i \in \mathbb{R}^{n \times 1}$ are the attention weights

- $\mathbf{1}_m \in \mathbb{R}^{1 \times m}$ is a row vector of ones of length $m$

- $\mathbf{1}_n \in \mathbb{R}^{n \times 1}$ is a column vector of ones of length $n$

- $\odot$ is element-wise multiplication

## 4 Experiments

### 4.1 Datasets

For the experiments, three different publicly available datasets were used in order to thoroughly test the models under numerous real-world scenarios, dataset sizes, and noise levels. The first two datasets were the "diabetes" and "California housing" datasets from the scikit-learn library (Pedregosa et al., 2011; noa, a;d). The last dataset was a synthetic dataset generated from the scikit-learn library's make_regression function. According to the scikit-learn documentation, the noise parameter is defined as "The standard deviation of the gaussian noise applied to the output" (Pedregosa et al., 2011; noa, e). Table 1 below shows the aspects of each dataset.

| Dataset | Samples | Features | Noise |
|---------|---------|----------|-------|
| Diabetes | 442 | 10 | NA |
| Synthetic | 5,000 | 10 | 5 |
| Housing | 20,640 | 8 | NA |

Table 1: Comparison of datasets used for model testing

### 4.2 Parameters

Each model was tuned for each dataset using a grid search technique (noa, b). The target of the grid search was to minimize the mean squared error (MSE) of each model on the testing split. The full ranges of the grid search and the total number of parameter combinations tested for each model across all datasets can be found in Appendix A. (Table 2). The optimal parameters can also be found in Appendix A. (Table 3). In total, 195,048 different combinations of parameters were tested across the datasets.

### 4.3 Metrics

Four different metrics were used to get a thorough evaluation of the performance of each model: mean squared error (MSE), mean absolute error (MAE), $R^2$, and root mean squared error (RMSE). MSE is the mean of the differences in real and predicted values squared. Since MSE squares the difference, it highlights larger errors. However, it is not in the same units as the data since it is squared. RMSE is the square root of the MSE, which aims to increase interpretability by setting the units to be the same as the target variable. MAE is the mean of the absolute value of the difference between the real and predicted values. It provides a more straightforward and interpretable measure of error with an equal weighting to all errors (Botchkarev, 2019). $R^2$ is a measure of the proportion of the variance of the target data that can be predicted using the model. In other words, the goodness of the fit of the model. Due to this, it is a strong and interpretable metric for the overall performance of a model. An $R^2$ of 1 is a perfect fit, while 0 means the model explains none of the variance, and negative means it performs worse than a horizontal line (the mean of the data) (Chicco et al., 2021).

| Model | MSE | MAE | $R^2$ | RMSE |
|---|---|---|---|---|
| XGBoost | 2781 | 42.26 | 0.4750 | 52.74 |
| LightGBM | 2728 | 40.54 | 0.4850 | 52.23 |
| Base Model | 2852 | 42.09 | 0.4618 | 53.40 |
| AMBeRBoost | 2795 | 43.24 | 0.4724 | 52.87 |

Table 2: Results From Diabetes Dataset

| Model | MSE | MAE | $R^2$ | RMSE |
|---|---|---|---|---|
| XGBoost | 106.8 | 7.864 | 0.9963 | 10.34 |
| LightGBM | 134.5 | 9.000 | 0.9954 | 11.60 |
| Base Model | 140.0 | 9.450 | 0.9952 | 11.83 |
| AMBeRBoost | 119.6 | 8.897 | 0.9959 | 10.94 |

Table 3: Results From Synthetic Dataset

## 5 Results

## 6 Discussion

The experiments show that the addition of the attention mechanism improves the performance of the base model on almost every metric. The only metric where AMBeRBoost performed more poorly than the base model is the MAE of the diabetes dataset. This implies that, although AMBeRBoost may have improved performance on noise (since the MSE and $R^2$ were still better), it has the risk of overfitting to small datasets. This finding supports the hypothesis that the addition of the attention layer can help the GBM understand noise. It is also observed that a similar effect happens with XGBoost—the base model had a higher MSE and $R^2$ but slightly lower MAE on the small diabetes dataset. For the synthetic dataset, which was medium size with relatively high noise levels, AMBeRBoost was able to achieve substantially higher results compared to the base model, even outperforming LightGBM. This result further highlights the efficacy of the attention layer on challenges with high levels of noise. Another notable advantage of AMBeRBoost is interpretability. The addition of the attention layer provides visual insights (see Figure 1 and Figure 2) into the learning and decision-making process that the model undergoes. Enhanced interpretability is very useful for improving and understanding model architectures, along with tuning hyperparameters.

## 7 Limitations and Future Directions

### 7.1 Grid Search

Despite grid search being a well-established and effective method for tuning hyperparameters, it has a number of limitations. First, by defining discrete values to be evaluated, it is likely that it is not considering the actual global minimum. Additionally, testing each combination of hyperparameters has a very high computational cost, which further limits the size of the hyperparameter space that can be searched. It also limits the number of hyperparameters that can be tested. Although a sizable number of hyperparameters for each model were tuned for this study, XGBoost (Chen and Guestrin, 2016) and LightGBM (Ke et al., 2017) still had a number of hyperparameters left as default. In practice, it is common to leave some hyperparameters as their default values, but it is possible that further fine-tuning may have additionally improved model performances. Lastly, overfitting is a risk of hyperparameter tuning since the results are tuned specifically for the single testing split. Future research should be done to contrast the models evaluated in this paper in larger or more complex hyperparameter spaces. More efficient hyperparameter tuning techniques, like Bayesian methods (Snoek et al., 2012), could be employed. Additionally, techniques like cross-validation could be utilized to lower the risk of overfitting (Berrar, 2018).

| Model | MSE | MAE | $R^2$ | RMSE |
|---|---|---|---|---|
| XGBoost | 0.1901 | 0.2859 | 0.8549 | 0.4361 |
| LightGBM | 0.1897 | 0.2861 | 0.8553 | 0.4355 |
| Base Model | 0.2051 | 0.2952 | 0.8435 | 0.4529 |
| AMBeRBoost | 0.2029 | 0.2939 | 0.8452 | 0.4504 |

Table 4: Results From California Housing Dataset

## 7.2 Datasets

No dataset is perfect. Although numerous datasets were tested in this study to attempt to get a more thorough analysis, they are still far from exhaustive. Future research should be conducted to test AMBeRBoost across more datasets with a larger variance of characteristics. Additionally, research should be done to test the effectiveness of AMBeRBoost in more real-world applications.

## 7.3 Base Model

The base model used for AMBeRBoost was a very simple GBM, which performed the worst on almost all of the performance metrics. Future research should test the effects of implementing the attention layer from AMBeRBoost into more advanced models, such as XGBoost (Chen and Guestrin, 2016) or LightGBM (Ke et al., 2017). Additionally, more sophisticated techniques, like regularization, could be implemented into the AMBeRBoost model to help reduce overfitting.

## 7.4 Attention Layer

Although the implementation of the attention layer was able to produce fairly strong results, it is still far from perfect. Future research should aim to improve the architecture of the network, increasing speed, predictive capabilities, and stability. Additionally, a more complex neural network may capture more complex relationships within the data. Integrating a separate training cycle where backpropagation (Rumelhart et al., 1986) is performed on the network may also yield improvements on the network's ability to capture more complex information relating to the data.

## 7.5 Classification

The implementation outlined in this paper was for a regression model. However, GBMs have shown great success with classification tasks as well (Rufo et al., 2021; Alzamzami et al., 2020; Cherif and Kortebi, 2019). Future research should test the effectiveness of implementing attention networks into the training of GBM classification models.

## 7.6 Hyperparameters

AMBeRBoost has two additional hyperparameters compared to the base model: the number of hidden units and influence of the attention weights (lambda). Further research should be done to make these hyperparameters more adaptable by default, possibly scaling by factors in the data, like its size. Hyperparameters like the learning rate and max depth could also be scaled to further reduce the complexity of hyperparameter tuning. Additionally, dynamically scaling hyperparameters during training may help reduce overfitting, increase adaptability, and further reduce the cost of hyperparameter tuning.

# 8 Conclusion

This study has introduced a novel implementation of an attention network into the training process of a GBM. Experiments were conducted to demonstrate that the introduction of the attention network can increase performance metrics across multiple varying datasets, particularly by bolstering its predictive capabilities with noise. These findings have important implications for the field of machine learning by bridging the

gap between neural attention mechanisms and GBMs and opening the door for several promising avenues of future research.

## A

| Parameter | XGBoost | LightGBM | Base Model | AMBeRBoost |
|---|---|---|---|---|
| number of combinations tested | 68,040 | 119,070 | 378 | 7,560 |
| n_estimators | 300, 400, 500 | 300, 400, 500 | 300, 400, 500 | 300, 400, 500 |
| max_depth | 1, 2, 3, 4, 5, 6 | 1, 2, 3, 4, 5, 6 | 1, 2, 3, 4, 5, 6 | 1, 2, 3, 4, 5, 6 |
| learning_rate | 0.1, 0.15, 0.2, 0.25, 0.3, 0.4, 0.5 | 0.1, 0.15, 0.2, 0.25, 0.3, 0.4, 0.5 | 0.1, 0.15, 0.2, 0.25, 0.3, 0.4, 0.5 | 0.1, 0.15, 0.2, 0.25, 0.3, 0.4, 0.5 |
| colsample_bytree | 0.5, 0.6, 0.7, 0.8, 0.9, 1.0 | NA | NA | NA |
| gamma | 0, 0.1, 0.2, 0.3, 0.4, 0.5 | NA | NA | NA |
| subsample | 0.6, 0.7, 0.8, 0.9, 1.0 | 0.6, 0.7, 0.8, 0.9, 1.0 | NA | NA |
| num_leaves | NA | 20, 30, 40, 50, 60, 70, 80, 90, 100 | NA | NA |
| min_child_samples | NA | 5, 10, 20, 30, 40, 50, 60 | NA | NA |
| hidden_units | NA | NA | NA | 128, 256, 512, 1024, 2048 |
| lambda | NA | NA | NA | 0.08, 0.09, 0.1, 0.11 |

Table 5: Range of Parameters Tested During Grid Search

| Parameter | XGBoost | LightGBM | Base Model | AMBeRBoost |
|---|---|---|---|---|
| n_estimators | 500, 300, 500 | 500, 300, 500 | 500, 300, 500 | 500, 500, 500 |
| max_depth | 2, 5, 6 | 1, 1, 6 | 1, 1, 6 | 1, 1, 6 |
| learning_rate | 0.4, 0.15, 0.1 | 0.15, 0.1, 0,1 | 0.15, 0.1, 0.1 | 0.15, 0.15, 0.1 |
| colsample_bytree | 0.5, 0.5, 0.8 | NA | NA | NA |
| gamma | 0, 0.5, 0 | NA | NA | NA |
| subsample | 0.7, 0.8, 0.7 | 0.6, 0.6, 0.6 | NA | NA |
| num_leaves | NA | 20, 20, 30 | NA | NA |
| min_child_samples | NA | 5, 30, 20 | NA | NA |
| hidden_units | NA | NA | NA | 1024, 1024, 128 |
| lambda | NA | NA | NA | 0.11, 0.11, 0.11 |

Table 6: Results of Grid Search For Synthetic, Diabetes, and Housing Datasets Respectively

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
