# OpenReview forum: "Enhancing Gradient Boosting Machines with Attention"
_TMLR — Rejected by TMLR_

### Review · Reviewer_Zme8 · 2024-12-19

**Summary Of Contributions:**

The authors propose to combine boosted decision trees with an attention mechanism. They then go on to test the proposed method on a few simple benchmarks.

**Audience:**

No

**Claims And Evidence:**

No

**Requested Changes:**

Overall, I’m afraid that, in my opinion, way too major modifications would be required to make this paper suitable for TMLR. Not only the method would need to be better explained, but also an important amount of development would be required to make the manuscript of interest to the TMLR readership.

**Strengths And Weaknesses:**

There are several major issues with this submission:

1) I could not grasp the actual nature of the method.

For instance, the explanation in Algorithm 2 is not precise enough to understand what it does. What are the gradients? They must be the partial derivative of some scalar with respect to something. But, to what? I should assume of some likelihood with respect to the outputs of some weak learners. But then, what does it mean to expand the gradients to match the dimension of the input? What does it mean to concatenate the inputs and the expanded gradients?

Similarly, I could not understand how attention can be computed sequentially as described in Algorithm 3, or what such attention would mean.

2) I am aware that TMLR does not strictly require methodological novelty, but I could not understand what is either the claimed novelty or interesting observation that the authors want to share with this article.

3) The reported results show that the proposed method performs comparably to the baseline (we would need to see the statistical significance of the metric differences) and substantially worse than other previous methods, which are all around 8 years old.

I am not an expert on gradient boosting models, and I apologize if there is something important I may have missed due to that.

---

### Review · Reviewer_cUaV · 2025-01-08

**Summary Of Contributions:**

This paper proposes AMBeRBoost, an attention mechanism-based regression gradient boosting machine (GBM). This enhances the ability of GBM to adapt to and capture complex patterns in the data.

**Audience:**

Yes

**Claims And Evidence:**

Yes

**Requested Changes:**

In addition to my questions above, I have a few more concerns.
- In Section 2.2, can you explain a bit more on how transformers help improve interpretability?
- In Algorithm 1, can you please include the description of what each notation means in the pseudocode itself?

**Strengths And Weaknesses:**

### Strengths
- The paper is mostly well-written, the experiment details are adequately explained.

### Weaknesses
**Inadequate explanations**
- In Algorithm 1, I did not fully understand what is happening in the for loop. Can you please explain the math behind it? Similar comment about Algorithm 3.
- In Figure 1 and 2, the difference seems to be the range of weights. Why does this happen? Why is it significant?
- What is happening in (1)? Why does it make sense?
- I did not understand the interpretability advantage of AMBeRBoost. How do the visual insights of Fig 1 and 2 help us?
- Why does XGBoost outperform AMBeRBoost on so many metrics across all datasets?

**Experiment results not adequate/comprehensive**
- I appreciate the authors pointing out the limitations of their work in Section 7. However, I also wonder why the authors did not experiment with classification in this paper itself. This is not a long paper, so it's not that space was a concern.

---

### Review · Reviewer_Wgyd · 2025-01-22

**Summary Of Contributions:**

The paper proposes to improve gradient boosting machine (GBM) by weighing the inputs of the weak learners using outputs of an attention layer.

The proposed method is called AMBeRBoost, which outperforms the attention-less base model but is worse than classic methods such as XGBoost and LightGBM.

**Audience:**

Yes

**Broader Impact Concerns:**

There is no ethical concerns.

**Claims And Evidence:**

No

**Requested Changes:**

- Please experiments with more datasets and stronger base models, which are currently mentioned as future work but I think are necessary to make the method convincing.
- Please specify implementation details; e.g. In Algorithm 2, what's the input and output dimensions? Are the operations performed element-wise or over all dimensions? Why using sigmoid rather than softmax? Perhaps using equations rather than plain English can help clarify.
- Clarification: For Table 1, what is the variance of the features themselves? i.e. How significant is the Gaussian noise with a standard deviation of 5?

**Strengths And Weaknesses:**

Strengths:
- The proposed AMBeRBoost improves over the attention-less baseline on high noise setups, e.g. a synthetic dataset with Gaussian noise.
- The paper discusses the necessary backgrounds, on both gradient boosting machines and attention.

Weaknesses:
- There is limited justification on the use of attention and the design choices.
- The current empirical results are not sufficient to demonstrate the effectiveness of the method: the results are on small-scale datasets only, and the gain of AMBeRBoost over the attention-less base model is small, and AMBeRBoost lags significantly behind XGBoost.
- The interpretability results in Fig 1 and 2 are minimal and provide little insights.
- The writing needs to be improved: for the algorithm boxes, please consider using more mathematical writing rather than plain English. Section 5 "Results" has no texts and is currently empty (perhaps because the result tables are placed in other places).

---

### Decision · Action_Editor_pGiW · 2025-03-11

**Recommendation:** Reject

**Comment:**

The topic of the paper (combining gradient boosting with attention) is interesting. However, as noted by the reviewers, the paper suffers from a few drawbacks. The method is not very clearly or precisely defined; the baselines are somewhat basic; and the evaluations are limited to small-scale test datasets.

For the paper to meet the bar for publication, I would recommend significantly improving the clarity of the methods section, selecting more challenging baselines for comparisons with state-of-the-art, and showing that the proposed method meets/exceeds the state-of-the-art on a modern tabular data regression benchmark such as TabZilla.

**Audience:**

The topic of the paper is interesting and within scope. However, the method could be more clearly stated. As such the results are hard to interpret and do not seem to provide actionable insights.

**Claims And Evidence:**

The claim that the proposed method meets/exceeds the state-of-the-art is not justified by the lack of sufficient baselines or the (very) toy experiments.

**Resubmission Of Major Revision:**

The authors may consider submitting a major revision at a later time.